# Mucosal Application of a Low-Energy Electron Inactivated Respiratory Syncytial Virus Vaccine Shows Protective Efficacy in an Animal Model

**DOI:** 10.3390/v15091846

**Published:** 2023-08-30

**Authors:** Valentina Eberlein, Mareike Ahrends, Lea Bayer, Julia Finkensieper, Joana Kira Besecke, Yaser Mansuroglu, Bastian Standfest, Franziska Lange, Simone Schopf, Martin Thoma, Jennifer Dressman, Christina Hesse, Sebastian Ulbert, Thomas Grunwald

**Affiliations:** 1Fraunhofer Institute for Cell Therapy and Immunology, 04103 Leipzig, Germany; valentina.eberlein@izi.fraunhofer.de (V.E.);; 2Fraunhofer Cluster of Excellence Immune-Mediated Diseases CIMD, 60596 Frankfurt am Main, Germanyyaser.mansuroglu@itmp.fraunhofer.de (Y.M.);; 3Fraunhofer Institute for Toxicology and Experimental Medicine, 30625 Hannover, Germany; 4Fraunhofer Institute for Organic Electronics, Electron Beam and Plasma Technology FEP, 01277 Dresden, Germany; 5Fraunhofer Institute for Translational Medicine and Pharmacology, 60596 Frankfurt, Germany; 6Fraunhofer Institute for Manufacturing Engineering and Automation, 70569 Stuttgart, Germany

**Keywords:** respiratory syncytial virus, RSV, mucosal vaccine, inactivated vaccine, low-energy electron irradiation, LEEI, PC formulation, PCLS

## Abstract

Respiratory syncytial virus (RSV) is a leading cause of acute lower respiratory tract infections in the elderly and in children, associated with pediatric hospitalizations. Recently, first vaccines have been approved for people over 60 years of age applied by intramuscular injection. However, a vaccination route via mucosal application holds great potential in the protection against respiratory pathogens like RSV. Mucosal vaccines induce local immune responses, resulting in a fast and efficient elimination of respiratory viruses after natural infection. Therefore, a low-energy electron irradiated RSV (LEEI-RSV) formulated with phosphatidylcholine-liposomes (PC-LEEI-RSV) was tested ex vivo in precision cut lung slices (PCLSs) for adverse effects. The immunogenicity and protective efficacy in vivo were analyzed in an RSV challenge model after intranasal vaccination using a homologous prime-boost immunization regimen. No side effects of PC-LEEI-RSV in PCLS and an efficient antibody induction in vivo could be observed. In contrast to unformulated LEEI-RSV, the mucosal vaccination of mice with PC formulated LEEI-RSV showed a statistically significant reduction in viral load after challenge. These results are a proof-of-principle for the use of LEEI-inactivated viruses formulated with liposomes to be administered intranasally to induce a mucosal immunity that could also be adapted for other respiratory viruses.

## 1. Introduction

Human respiratory syncytial virus (RSV) is a highly infectious and seasonally occurring member of the Pneumoviridae family that can lead to upper and lower respiratory tract infections (LRTI) [1,2,3]. Patients such as infants, especially in the first six months of life, pre-term born, the elderly over 60 years of age, or patients suffering from additional lung pathologies are at high risk of severe lung disease after RSV infection [1,2,4]. In 2019, the global burden of RSV was approximately 33 million associated LRTI and 101,400 RSV-attributed deaths in children under six years of age [4]. In the elderly, RSV has a similar or even greater burden than influenza, as evidenced in prolonged hospital stays, more intensive care unit admissions, and higher mortality [5,6,7,8].

The vaccine development against RSV has been faced with several drawbacks. In the 1960s, a formalin-inactivated RSV (FI-RSV) vaccine trial in children enhanced the severity of the disease after natural reinfection, which was associated with bronchopneumonia, pulmonary eosinophilia, and extensive monocyte infiltrations, with two fatal cases [9,10].

Since 1986, passive immunization with the monoclonal antibody palivizumab [1,11,12,13], and more recently, the improved nirsevimab, are available [14,15]. Several vaccine approaches have currently entered clinical trials and the first vaccines were approved by the Food and Drug Administration (FDA) in 2023 [8,16,17]. Two recombinant subunit vaccines containing stabilized RSV prefusion-F protein have been FDA-approved for adults 60 years and above. The immunizations showed an efficacy against RSV-related LRTI of 82.6% (AReSVi-006) and 85.7% (RSVPreF), respectively [17,18,19,20,21,22]. The latter prefusion F protein vaccine (RSVPreF) approved for older adults (NCT05035212) also showed protection in infants after maternal vaccination (NCT04424316) [16,21,22,23,24,25]. In addition, an mRNA-vaccine expressing the stabilized RSV pre-F is currently in phase III clinical trial [26,27]. Further vaccine approaches include vector-based vaccines such as MVA-BN-RSV, a vector expressing the F, G, M2, and N protein of RSV in phase III [8,27], and live-attenuated vaccines, virus-like particles, and nanoparticles [8,27].

Despite these great breakthroughs, all setups are based on intramuscular applications whereby a mucosal vaccination route could be beneficial against respiratory viruses. The mucosal application is atraumatic and may enhance the vaccination acceptance in the community over needle-based applications [28,29]. Most importantly, mucosal vaccinations against respiratory viruses trigger the mucosal immune system, promoting local immune responses, resulting in a fast and efficient elimination of viruses directly in the respiratory tract [28,30]. The induction of the mucosal immunity can be advantageous over systemic vaccination approaches, especially in RSV-naïve children, as the latter could lead to the overwhelming immune pathology known after natural RSV infection [31,32]. Only a few vaccine candidates have the potential to be safe and effective after mucosal application. In particular, vector vaccines and live-attenuated vaccines are known candidates, as promising results in preclinical trials have been shown and may have favorable outcomes for patients [30,33,34]. Even though live-attenuated and vector vaccines have great potential, reactogenicity, reversion to a virulent pathogen, or the possibility of retrograde transport into the brain, are undesirable risks [35,36,37]. Inactivated vaccines can circumvent these biosafety risks, since pathogens are no longer capable of the replication in the vaccinees.

Virus inactivation approaches, especially those using physical methods, are safe and have low production costs [38,39,40]. We have shown previously that low-energy electron irradiation (LEEI) is a safe, non-toxic, and non-probe harming inactivation method [41,42,43,44,45,46]. The advantage over other irradiation methods is that the emission of secondary photon radiation is minimal, reducing the need for extensive shielding and making LEEI-technologies applicable in standard laboratories [43,47]. Since the penetration depth of low-energy electrons is highly limited [48,49], we have developed automated processes that generate thin liquid films, enabling the efficient LEEI of pathogens in suspension up to multi-liter scales [41,42,43,46]. LEEI has advantages over other radiation types such as ultraviolet (UV) light. It was previously shown that UV light is harmful to viral proteins including RSV-F, the most relevant protein for the induction of protective immune responses against RSV [50,51]. This is in line with the observation that RSV-F is also damaged by treating RSV with formalin [52,53]. It has been shown that this misfolding of surface RSV-F proteins was one major reason for the unfortunate outcome of the vaccine trial in the 1960s, resulting in poor protective [10,53,54] and unbalanced Th2 immune responses [55,56].

In contrast to these studies with misfolded RSV-F protein in formalin-inactivated RSV vaccine preparations, we have previously shown that in LEEI-inactivated RSV (LEEI-RSV), at least 70% of the RSV-protein is present in the Pre-F-conformation [44]. In the associated preclinical trial, intramuscular application of LEEI-RSV with Alhdydrogel showed high protection upon RSV challenge, which was associated with no significant immune pathology in the lungs of infected mice [44].

In the present study, we investigated the mucosal application of LEEI-RSV for the first time. As the mucosa forms a physical and enzymatic barrier, inactivated vaccines applied mucosally should be formulated for resorption and immune stimulatory reasons [28]. LEEI-RSV was formulated with liposomes as they are known to be safe, low, or non-toxic and easy to produce [29,57]. As a liposome, phosphatidylcholine (PC) was chosen to provide a broad mucosal and systemic immune response after instillation [58,59,60,61].

RSV is one main reason for hospitalizations in infants and the elderly. Even though first vaccines directed against RSV-F have been approved, a mucosal whole virus vaccine could be beneficial with a broader immune response, especially in combinational use with the approved vaccines. Therefore, we developed an inactivated vaccine using LEEI and tested it in a mucosal vaccination model. The formulated PC-LEEI-RSV showed no safety concerns in the ex vivo murine precision cut lung slice (PCLS) model. The intranasal application of PC-LEEI-RSV was well-tolerated and reduced the viral load in mice. Our results indicate that the formulation of LEEI inactivated RSV mediates protective efficacy after mucosal homologous prime and boost vaccination.

## 2. Materials and Methods

Cell Culture and Virus Production

Type 2 human epithelial cells (HEp-2; ATCC, Manassas, VA, USA) were used for all RSV production and in vitro assays. Cells were maintained in Dulbecco’s modified Eagle’s medium (DMEM) with GlutaMAX (Thermo Fisher Scientific, Dreieich, Germany), containing 10% heat inactivated fetal calf serum (FCS) and antibiotics (100 IU/mL penicillin with 100 µg/mL Streptomycin, Thermo Fisher Scientific, Dreieich, Germany) at 37 °C with 5% CO_2_.

RSV laboratory strain A long was obtained from ATCC (VR-26). M. Peeples and P. Collins (NIH, Bethesda, MD, USA) kindly provided the recombinant RSV expressing GFP (rgRSV). Virus propagation and titer determination was performed as previously described [34,62]. In short, the different virus strains were propagated on 90% confluent monolayers of HEp-2 cells. Cells were infected with a multiplicity of infection (MOI) of 0.1 in FCS-free medium for 3 h at 37 °C with 5% CO_2_. Unbound virus was removed by replacing the medium with fresh 1% FCS containing medium and incubated for 72 h at 37 °C with 5% CO_2_. The infected cell supernatant was clarified by centrifugation for 5 min at 2000× *g* and 4 °C, followed by filtration through a 0.45 µm-filter and ultracentrifugation at 21,000× *g* through a 20% (*w*/*v*) sucrose cushion in PBS for 3 h at 4 °C in a SureSpin 630 swing-out rotor (Thermo Fisher Scientific, Dreieich, Germany). The pelleted virus was resuspended in 10% (*w*/*v*) sucrose in PBS and titrated using a focus forming assay or a tissue culture infectious dose 50 (TCID50) assay. For the focus forming assay, confluent monolayers of HEp-2 cells were infected with serial dilutions of the virus and incubated for 48 h at 37 °C with 5% CO_2_. Viral rgRSV foci were visualized by fluorescence and RSV-long was analyzed by immunocytochemical (ICC) staining with an anti-RSV antibody (AB1128; Sigma Aldrich, Taufkirchen, Germany) [63]. For the TCID50-assay, viral stocks were diluted in 10-fold increments and incubated on confluent HEp-2 cell monolayers in a 96-well microwell plate for a period of five to six days. The cells were monitored for cytopathic effects (CPE) and the titer was calculated using the Reed–Muench method [64].

2.Virus Inactivation

a.Low-energy electron irradiation (LEEI)

RSV samples were irradiated in a custom-built irradiation device situated in a BSL2 laboratory at the Fraunhofer Institute for Cell Therapy and Immunology [41]. This can be equipped with different modules constructed as research prototypes to enable the automated LEEI of liquid samples. RSV was inactivated in a system using disposable bags. Briefly, bags were filled with 10 mL each of an RSV-solution diluted in PBS with 10% (*w*/*v*) sucrose and sealed. Based on previous experiences, the samples were treated with LEEI of 300 keV, 1.2 mA [41,44]. Controls underwent the same process without applying LEEI. Afterward, the bags were reopened and the liquid collected for further testing. Inactivation of RSV was confirmed in a cell culture assay as described previously [41]. In short, HEp-2 cells in a 6-well cell culture plate were inoculated with 100 µL per well of irradiated samples and controls. The development of cytopathic effects (CPE) was monitored for five to six days before the supernatant was passaged to fresh cells. After an additional five to six days of incubation, the samples were considered inactivated when no CPE was visible.

b.Dosimetric analysis

The absorbed dose applied was estimated by using a liquid dosimeter based on 2,3,5-triphenyl-tetrazolium chloride (TTC, Carl Roth, Karlsruhe, Germany), which undergoes colorization due to a dose-dependent reaction to red formazan, as described in the study of Schopf et al. [65]. Previously, a calibration function to convert absorbance values into dose was found with a combined uncertainty of 11.8% in the dose range of 6.5–38 kGy. That calibration was performed at the radiation plant REAMODE of Fraunhofer FEP using a standard reference film dosimeter Risø B3 (DTU Health Tech, Lyngby, Denmark).

The TTC was filled in the bag and irradiated with a constant acceleration voltage of 300 kV. The beam current was varied from 0 to 2.0 mA in intervals of 0.5 mA, and a linear regression was conducted to calculate the resulting dose in the unit gray (Gy).

3.ELISA RSV Conservation after LEEI

To examine the conservation of the antigenicity of RSV, enzyme-linked immunosorbent assays (ELISAs) were performed as previously described [41,44]. Briefly, 5 µL irradiated RSV samples and controls were coated on black NUNC 96-well MicroWell™ PolySorp^®^ plates (Thermo Fisher Scientific, Dreieich, Germany) in carbonate coating buffer (35 mM NaHCO_3_ (Carl Roth, Karlsruhe, Germany), 15 mM Na_2_CO_3_ (Carl Roth, Karlsruhe, Germany), pH 9.6) in a total volume of 100 µL/well overnight at 4 °C. For a quantitative standard curve, RSV was coated in fivefold dilutions in concentrations ranging from 5 × 10^2^ to 1 × 10^5^ FFU and processed as the LEEI-RSV. To convert the FFU values to TCID50, the correlation factor 0.7 was used [66]. The plate was washed three times with PBS containing 0.05% Tween 20 (PBS-T, Carl Roth, Karlsruhe, Germany) and blocked with 5% (*w*/*v*) skimmed milk powder (Carl Roth, Karlsruhe, Germany) in PBS (Bio&Sell, Feucht, Germany) for 2 h at room temperature. A monoclonal antibody recognizing RSV-F (18F12, [62]), diluted 1:200 in 2% (*w*/*v*) skim milk in PBS, was added and incubated for 2 h at room temperature, followed by the polyclonal Peroxidase AffiniPure Sheep Anti-Mouse IgG (H + L) antibody (Dianova, Hamburg, Germany) diluted 1:500. After 1 h of incubation, the readout was performed using a Centro XS3 luminometer (Berthold Technologies, Bad Wildbad, Germany). Enhanced chemiluminescent substrate (ECL, Pierce, Waltham, MA, USA) was diluted 1:10 in PBS and 100 µL of the substrate was injected into each well after 1.5 s of delay. Relative light units (RLU) were counted for 1 s.

4.Lipid Production and Virus Packaging

Phosphatidycholine (PC, LIPOID GmbH, Ludwigshafen, Germany) liposome formulations were prepared by a thin-film hydration method, followed by size reduction via manual extrusion. Forty mg of lipids was dissolved in 1.5 mL methanol (VWR, Darmstadt, Germany) and evaporated in a rotavapor (Büchi R-114, Büchi Labortechnik AG, Flawil, Schweiz) by reducing the pressure to 500 mPa, followed by further reductions to 150 mPa and finally to 50 mPa. The pressure was held at each of these values for 30 min before proceeding to the next pressure reduction. The whole process was conducted at 37 °C. The resulting lipid film was rehydrated by adding 1 mL of PBS pH 7.4 buffer (Fisher Scientific GmbH, Schwerte, Germany) and vortexing for 20 min. The liposome mixture was then extruded manually through a 200-nm polycarbonate membrane (Avestin, Mannheim, Germany) using a two-chamber manual extruder (Avestin, Mannheim, Germany). This procedure resulted in unilamellar liposomes. In order to prevent the loss of activity, the vaccine was incorporated into the vesicles after manufacturing the liposome formulation. The inactivated virus material was mixed in a ratio of 1:5 (liposomes:LEEI-RSV) with the liposome formulation and vortexed for 5 min. This material was kept at 4 °C prior to testing.

5.Precision Cut Lung Slices (PCLSs)

a.Preparation and treatment of murine, precision cut lung slices (PCLSs)

Female BALB/c mice (Charles River Laboratories, Sulzfeld, Germany) were sacrificed at three months of age. Lungs were resected and filled with a warm 2% agarose (Sigma Aldrich, Taufkirchen, Germany) solution in DMEM/F12 (Gibco, Waltham, MA, USA). Solidified lung lobes were cut into slices with a 300 µm thickness on a vibratome (Krumdieck Tissue Slicer, Alabama Research and Development, Muniford, AL, USA) in EBSS (Th Geyer, Renningen, Germany). The generated murine precision cut lung slices were treated with different concentrations of the vaccine candidates and cultured in DMEM/F12 supplemented with penicillin and streptomycin (10,000 U/mL, Gibco, Waltham, MA, USA) at 37 °C with 5% CO_2_ for 24 h.

b.Viability testing

To assess the viability of the murine precision cut lung slices (mPCLSs), the amount of released lactate dehydrogenase (LDH) was analyzed using the Cytotoxicity Detection Kit (Roche, Basel, Switzerland). The metabolic activity was measured using the Cell Proliferation Reagent WST-1 (Roche, Basel, Switzerland). Furthermore, cells were marked using the LIVE/DEAD Viability/Cytotoxicity Kit (Invitrogen, Waltham, MA, USA) to visualize the viable and dead cells by confocal laser scanning microscopy (LSM 800, Zeiss, Jena, Germany). All kits and reagents were applied at the manufacturer’s recommendations.

c.Cytokine secretion

To assume the immune response of the vaccine-treated mPCLS, the concentration of TNF-α was measured in the supernatants by ELISA (RnD, DuoSet, Minneapolis, MN, USA). Furthermore, released IFN-α, -β, -γ, IL-1β, IL-6, IL-10, IP-10, KC, MCP-1, MIP-1α, and RANTES were quantified by the U-Plex assay (Mesoscale Discovery, Rockville, MD, USA). All assays were performed following the manufacturer’s recommendations at appropriate sample dilutions.

6.Immunization and RSV Challenge in Mice

Female BALB/c mice were purchased at Charles River Laboratories (Sulzfeld, Germany) or breed at the Center for Experimental Medicine at the Fraunhofer Institute of Cell Therapy and Immunology and maintained under a specific pathogen-free environment in isolated ventilated cages. Seven- to eight-week-old mice were included in the experiment. Animal experiments were carried out according to the EU Directive 2010/63/EU for animal experiments and were approved by local authorities. Groups of five mice each were vaccinated in a homologous prime-boost manner at a four-week regime. Vaccination was intramuscular (i.m.) with LEEI-RSV adjuvanted with Alhdydrogel (LEEI-RSV i.m.) with 50 µL per hind leg (3 × 10^6^ TCID50/mL) or intranasal (i.n.) with PC-formulated LEEI-RSV (2 × 10^5^ TCID50/mL) or non-formulated LEEI-RSV with 40 µL per immunization. Blood for serum samples was collected one week before the first vaccination and three weeks after prime and boost vaccination. Mice were challenged four weeks after boost immunization with 10^6^ FFU RSV per animal intranasally as previously published [44]. Five days after infection, mice were euthanized and lungs were isolated for analysis of the viral load. In detail, lungs were homogenized in gentlMACS^TM^ M Tubes (Miltenyi Biotec., Bergisch Gladbach, Germany) with 2 mL ice-cold PBS using a gentlMACS Dissociator (Miltenyi Biotec., Bergisch Gladbach, Germany). After centrifugation at 2000× *g* for 5 min at 4 °C, the cleared supernatant was stored at −80 °C before viral RNA isolation [63].

7.RSV RNA Copy Analysis with qRT-PCR

Viral RNA was isolated from 140 µL of the cell-free lung homogenate supernatant using the QIAamp-Viral-RNA-Mini-Kit (Qiagen, Hilden, Germany) according to the manufacturer’s instructions. To determine the RSV-copy numbers, 45 ng of RNA was reverse transcribed and analyzed with the QIAGEN QuantiTECT RT-qPCR Kit using the RSV sense primer (5′-AGATCAACTTCTGTCATCCAGCAA-3′), RSV antisense primer (5′-GCACATCATAATTAGGAGTATCAAT-3′), and SYBR Green for detection in LightCycler^®^ 480 (Roche, Basel, Switzerland). Synthetic RSV-RNA of T7-transcripts served as the standard for the quantification of viral genome copy numbers [34].

8.Analyzing RSV-Specific Neutralizing Antibodies

RSV neutralizing antibody titers in the sera were determined by co-incubation with rgRSV. After sixfold dilution of the mouse sera in Hanks’ balanced salt solution (HBSS, Thermo Fisher Scientific, Dreieich, Germany), the complement was inactivated by incubation at 56 °C for 30 min. Serial twofold dilutions in DMEM with GlutaMAX (Thermo Fisher Scientific, Dreieich, Germany) containing 10% FBS and penicillin/streptomycin (Thermo Fisher Scientific, Dreieich, Germany) or a negative control without serum was incubated with around 100 FFU rgRSV per well for 1 h at 37 °C at 5% CO_2_. After 1 h, the serum containing virus was transferred on pre-seeded Hep2-cells and incubated for 48 h at 37 °C with 5% CO_2_. The analysis was performed by counting the green fluorescent viral foci using a FlouroSpot reader (AID Diagnistika, Straßberg, Germany). The neutralizing-antibody titer was defined as the highest serum dilution inhibiting rgRSV infection by more than 50% in comparison to the negative control (PRNT50 = 50% plaque neutralization titers). The detection limit of the neutralizing antibody was set at the lowest serum dilution (1:6) [34,44].

9.Analysis of RSV-Binding Antibodies in Mouse Sera

To examine the amount of RSV-binding IgG-antibodies in the sera of vaccinated animals, ELISA analyses were performed as previously described [41,44]. Briefly, purified and heat-inactivated (56 °C, 30 min) RSV was coated at a concentration of 5 × 10^5^ FFU/well on black NUNC 96-well MicroWell™ PolySorp^®^ plates (Thermo Fisher Scientific, Dreieich, Germany) in carbonate coating buffer (35 mM NaHCO_3_, 15 mM Na_2_CO_3_, pH 9.6) in a total volume of 100 µL/well at 4 °C for 24 h. The plate was washed three times with PBS containing 0.05% Tween 20 (PBS-T) and blocked with 5% (*w*/*v*) milk powder in PBS for 1 h at room temperature. The sera were diluted 1:2000 in 2% (*w*/*v*) skim milk in PBS and added in 100 µL to each well. After incubation for 2 h at room temperature, secondary antibodies for the total IgG, horse radish peroxidase (HRP) conjugated anti-mouse IgG antibody (Peroxidase AffiniPure Sheep Anti-Mouse IgG (H + L), polyclonal, Dianova, Hamburg, Germany) was added at a 1:1000 dilution and incubated for 1 h at room temperature. The readout was performed using enhanced chemiluminescent substrate (ECL, Pierce, Waltham, MA, USA) diluted 1:10 in PBS and 100 µL of the substrate was injected into each well. After 1.5 s of delay, the relative light units (RLU) were counted for 1 s at the Centro XS3 luminometer (Berthold Technologies, Bad Wildbad, Germany).

10.Statistical Analysis

Statistical analysis was performed using GraphPad Prism Version 6.07. Data were analyzed using the Mann–Whitney U-test. Level of statistical significance is indicated as follows: *: *p* ≤ 0.05, **: *p* ≤ 0.01, ***: *p* ≤ 0.001.

## 3. Results

### 3.1. LEEI Inactivation and Formulation of RSV

For pathogen inactivation, LEEI is a potent method, which we described earlier in detail [41]. For the vaccination of mice, we inactivated RSV with LEEI, therefore confirming the dosimetry and success of the inactivation process. LEEI-RSV was then formulated with PC for further testing.

#### 3.1.1. LEEI Inactivation in the Bag Module Leads to Sufficient Surface Conservation

To determine the current to use for an irradiation with 25 kGy, which we had already defined as a safe inactivation dose of RSV [44], a dosimetry of the custom-built prototype with the use of the bag module was performed. Therefore, the radiochromic liquid dosimeter TTC was used for an estimation of the applied dose [41]. Through an interpolation of the applied beam current (Figure 1a), a dose of 25 kGy was defined at 1.2 mA

The inactivation with LEEI was performed as described in the Methods section After the process, cells were infected with the LEEI-treated RSV-samples to verify inactivation. After the second passage, no CPE was detectable in the cells infected with RSV treated with 25 kGy of LEEI, thus being considered inactivated (Appendix A). This led to further testing of the material and analysis of the conservation of the F-glycoprotein. Hence, an ELISA was performed where the whole RSV-F-content was tested with 18F12 (Figure 1a). As a control, the processed material without irradiation (0 kGy) was tested and there was a non-statistically significant 1.4-fold reduction in the 25 kGy group (Figure 1b). To determine the RSV amount corresponding to the measured relative light unit (RLU) signal, a standard curve with a serial dilution of RSV was generated by ELISA (Appendix A). The amount of RSV was calculated for the processed material using this standard curve. For the process control (0 kGy), 30,276 RLU/s corresponded to 6.97 × 10^4^ TCID_50_ and for LEEI-RSV irradiated with 25 kGy, 21,392 RLU/s correlated with 4.58 × 10^4^ TCID50 (Figure 1b).

#### 3.1.2. Formulation of Inactivated Material

To apply inactivated RSV intranasally, the material was formulated in liposomes after inactivation. A solution of inactivated LEEI-RSV at a concentration of 2 × 10^5^ TCID50/mL in PBS with 10% (*w*/*v*) sucrose was added to the unilamellar liposome formulation at a LEEI RSV:lipid ratio of 5:1. Zetasizer measurements showed a size distribution of 200 nm ± 49 nm. The rather large variation in liposomal size may be a result of the known variation in particle size of the RSV itself, which ranges between 150 and 250 nm [67]. The RSV content of the formulated liposomes was verified by ELISA.

### 3.2. Evaluation of Adverse and Immunogenic Effects of the New Vaccine Ex Vivo

An easy and fast procedure to test for adverse effects or the acute toxicity of drugs and vaccines on lung tissues is the human or rodent precision cut lung slice (PCLS) technology [68,69,70]. Freshly prepared living murine lung tissue was sliced (300 µm thickness) into cell culture dishes and incubated with the two compounds for 24 h. In addition to the toxicity, the secretion of cytokines was measured.

The non-formulated and formulated inactivated RSV material, LEEI-RSV and PC-LEEI-RSV, respectively, was tested on murine PCLS for adverse effects. The slices were incubated with tenfold serial dilutions of the material ranging from 10^3^ to 10^6^ TCID50/mL for 24 h. The LDH-release as a parameter for the cytotoxicity was measured in the supernatants. The background level of LDH-release in the medium control (TCID50 = 0) was around 30–45% of the Triton-lysed cells (Figure 2a). For both vaccines, up to a concentration of 10^5^ TCID50/mL, no LDH release was detectable compared to the medium control (Figure 2a). In contrast, a dose of 10^6^ TCID50/mL PC-LEEI-RSV induced a median LDH release of 80% compared to 45% in the medium control, while with LEEI-RSV, the induction was only to 39% compared to 31% (Figure 2a).

Looking at cell proliferation via the WST-conversion, a similar picture was visible: at a dose of 10^5^ TCID50/mL, no changes compared to the medium control were visible (Appendix A). Subsequently, at a concentration of 10^6^ TCID50/mL, both LEEI-RSV and PC-LEEI-RSV showed strongly reduced WST-conversion (Appendix A).

Analysis of the toxic effects in a live/dead staining supported the findings that a dose of 10^5^ TCID50/mL PC-LEEI-RSV or LEEI-RSV showed no toxic effects on the lung tissue compared to only culture medium (mock) (Figure 2b).

Aside from the absence of adverse effects by the inactivated material, the secretion of cytokines was measured for PC-LEEI-RSV in a concentration of 10^4^ TCID50/mL to ensure that the measured effects were not induced as by-products of toxicity but by the vaccine itself. The measured amounts of the indicated cytokines showed no significant differences between the PC LEEI-RSV in comparison to the medium control after 24 h (Figure 2c). Therefore, we concluded that PC-LEEI-RSV does not induce any of the measured cytokines in 24 h. In addition, IFN-alpha, -beta, and -gamma were measured but were under the limit of detection of 25 pg/mL, 1.5 pg/mL, and 0.9 pg/mL, respectively, in both conditions.

### 3.3. PC-LEEI-RSV Induces Immune Responses and Protection in Mice after Vaccination

To test the immunogenicity and protective efficacy of PC-LEEI-RSV, we used a well-established in vivo infection model of RSV. BALB/c mice were mucosally vaccinated in a homologous prime-boost regimen to analyze RSV specific humoral immune responses and protection against an RSV challenge. Routine follow-up of all animals showed no side effects at any timepoint after the applications of the different vaccines.

#### 3.3.1. Humoral Systemic Immune Response after Vaccination

The induction of a humoral immune response was measured by analyzing serum samples from the blood of animals before vaccination (pre immune) and three weeks after prime and boost immunization, respectively.

Sera were used to analyze the systemic RSV neutralizing and RSV binding antibodies (Figure 3). RSV neutralizing antibodies in the unvaccinated control group were at the baseline level (Figure 3a). After prime and boost immunization, two out of five animals showed the induction of neutralizing antibodies after intranasal vaccination of PC-LEEI-RSV. The control group with intramuscular vaccination of Alhydrogel-adjuvanted LEEI-RSV showed a significant induction of RSV-neutralizing antibodies after boost vaccination (Figure 3a).

All intranasally vaccinated PC-LEEI-RSV animals produced RSV-specific IgG antibodies (Figure 3b). The induction was statistically significant in comparison to the unvaccinated animals and compared to the pre-immune levels of these mice (Figure 3b). The LEEI-RSV i.m. group showed a 16-fold higher amount of IgG antibodies than the PC-LEEI-RSV group. In both vaccinated groups, the induction was statistically significant in comparison to the unvaccinated animals and to the respective pre-immune levels of the mice (Figure 3b). Untreated animals only had baseline levels of the antibodies (Figure 3b).

#### 3.3.2. PC-ELLI-RSV Protects Mice after RSV Challenge

To test the induced protective efficacy, an RSV challenge experiment with 10^6^ FFU per mouse was performed. Animals were scored daily, and no clinical symptoms were detectable after RSV infection. Five days after challenge, mice were euthanized and the viral load was determined in the lungs.

PC-LEEI-RSV, given intranasally by a homologous prime boost vaccination regimen containing 8 × 10^3^ RSV particles per application, induced an immune response, which led to a 171-fold reduction in the viral load compared to the untreated animals (Figure 4). The viral load in LEEI-RSV i.m. mice was 966-fold lower than in the unvaccinated group, and therefore, the i.m. vaccination led to a 5.7-fold better reduction in the viral load compared to the intranasally applied PC-LEEI-RSV (Figure 4).

In another animal trial, the vaccination of mice intranasally with LEEI-RSV, without adjuvantation, did not show the induction of a sufficient protective immune response (Appendix A). Low titers of RSV-binding IgG antibodies were induced (Appendix A), and only a threefold reduction in viral load after RSV challenge was detectable (Appendix A). This shows that the PC-formulation is critical for the immunogenicity.

Since we also could not detect any sign of enhanced disease severity shown by weight loss or disease development in mice, we can confirm that LEEI-RSV is a safe and highly efficacious vaccine. In addition, by formulation with PC, this vaccine candidate can be applied intranasally without any further adjuvants.

## 4. Discussion

Mucosal vaccines induce protection at the site of infection by activating local immune responses [29,30,71,72,73].

In this study, we showed, for the first time, protection with the intranasal application of a PC-formulated LEEI-RSV in mice against RSV. To exclude adverse effects on viability or immune activation in the lung tissue, we first showed in the PCLS model that no toxic effects were generated with our vaccines. In the analysis, neither the LEEI-RSV itself nor the formulated PC-LEEI-RSV induced adverse effects in the murine lung tissue. The viability staining showed good viability with the vaccine candidates, and the WST and LDH only showed adverse results at the highest concentration tested. Importantly, the PC-LEEI-RSV vaccine induced systemic RSV-binding antibodies and serum neutralizing antibodies after mucosal vaccination. The detected levels of neutralizing antibodies close to the baseline in the unvaccinated boost-sera and the PC-LEEI-RSV pre immune sera were more likely unspecific background. Upon RSV challenge, a statistically significant protection in viral load in the lungs of vaccinated animals compared to unvaccinated mice was observed. We could thus demonstrate that the intranasal vaccination provides a reduction in the viral load, which might be sufficient to protect against RSV infection if LRTIs are blocked [74].

These encouraging results demonstrate, to our knowledge, a proof-of-principle of a novel mucosal applied vaccine candidate that is inactivated by LEEI and formulated to improve immunogenicity and protective efficacy. It is worth mentioning that the liposome-based formulation and the mucosal application can be further improved. For intranasal application, sophisticated devices have been developed to produce size-specific droplets. However, the optimization of this mucosal vaccine candidate was beyond this proof-of-concept study. The application route of LEEI inactivated vaccines has so far been the intramuscular route, but in this proof-of-concept study, we observed that the intranasal route also induced protective efficacy based on the reduction in the viral load. Nevertheless, the mucosal route needs the liposomal based formulation for the efficient induction of a protective immune response. Liposomes, especially PC, are physiological substances that are known to be non-toxic and are already medically applied [59]. Liposomes are widely used for drug delivery across a variety of therapeutic areas [58,59,60]. Many studies have already proven the non-toxicity of liposomes [75,76]; especially PC, as a physiological substance should not interfere with the respiratory system as a high proportion of the pulmonary surfactant consists of lipids [61,77,78].

Mucosal vaccines are to date mostly based on live-attenuated viruses (LAAV) and viral vectors as these vaccine candidates can naturally infect mucosal cells and overcome the mucosal barriers. LAAVs activate multiple immune responses including the innate immune system, mimicking a normal virus infection [8,72]. However, an RSV-LAAV still induced unwanted disease symptoms after application because it was insufficiently attenuated [79]. In contrast, over-attenuated RSV-LAAV failed to induce sufficient protection [80].

Viral vectors present another promising approach for mucosal application as they infect mucosal cells and express an immune relevant transgene that induces an immune response. Adenoviral vectors have shown great potential as a mucosal immunization platform, as protection was induced compared to an intramuscular application after a vaccination with the same doses [31,34]. Furthermore, the adenoviral vector vaccine offers higher protection against RSV compared to natural infection in mice including better humoral and cellular immune response at the site of infection [30]. We have previously described that an intramuscular prime with a DNA-plasmid encoding RSV-F, followed by a mucosal adenoviral vector boost, induces high amounts of mucosal T-cells, systemic humoral responses, and protective efficacy upon RSV challenge [81]. With regard to intramuscular prime and mucosal boost, we have recently shown that an adenoviral vector against SARS-CoV-2 induces high protective efficacy and superior mucosal immune responses in comparison to a homologous vaccination using the formulated mRNA intramuscular vaccine [82]. Aside from adenoviral vectors, another vector platform based on the Modified Vaccinia Ankara virus (MVA) is currently in clinical trials against RSV using the intramuscular approach [8]. A similar MVA-based vaccine was recently tested using the intranasal route, showing good antibody induction and protection against RSV in mice [83]. One larger disadvantage of virus vector vaccines is the induced vector immunity [84,85]. In both live attenuated viruses and virus vectors, it is important that no immune response against the vaccines with IgA or cytotoxic T-cells is already established because it could block the vaccine [84,85,86,87]. In addition, new concerns against the adenoviral vector vaccines arose in the context of vaccines against SARS-CoV-2 due to vaccine-induced immune thrombotic thrombocytopenia [88,89].

The issues of vector immunity and safety concerns with LAAV or vector vaccines can be circumvented by using inactivated vaccines [90]. Inactivated vaccines are not able to replicate and are mostly applied intramuscularly with different adjuvants to enhance and prolong the immune response [91,92]. For respiratory viruses such as RSV, the site of infection is the respiratory tract, and a mucosal vaccination would induce higher mucosal immunity [28,30,34,57,93]. Furthermore, it has already been observed that the induction of immune responses via a mucosal application include broad mucosal immune response and systemic immunity [30,31,93]. The intranasal application of an adenoviral vector vaccine against RSV led to the induction of RSV-F specific CD8+ T cells, central memory CD8+ T cells, and most importantly, to tissue-resident memory CD8+ T cells in the lungs of vaccinated mice [30]. Similar observations were seen with an intranasal RSV-F Nanovaccine inducing tissue-resident memory CD4+ and CD8+ T cells besides neutralizing antibodies [93]. These cellular immune response mechanisms are important and necessary in a balanced and protective immunity against RSV [94]. However, the analysis of cellular immune responses was beyond the scope of this proof-of-principle study. Besides the beneficial immunological aspects, a needle-free application of a vaccine might also reduce vaccine hesitancy, as it is a non-traumatic application method [28]. Additionally, the application is more feasible than an injection and the immunization can likely be performed without medical assistance, enhancing the performance of global immunization [28].

For the effective induction of an immune response after mucosal application, the vaccine has to overcome the physical and biological barriers of the mucosa, namely the cell layer with tight junctions and the mucus with proteoglycans, lipids, or DNA [71]. To overcome elimination by the mucosal defense mechanism, resorption, and immune stimulatory reasons, adjuvanting or packaging of the vaccine is necessary, whereby the choice of substance can be essential. For example, in an intranasal influenza vaccine, the use of the mLT LTK3 adjuvant led to transient peripheral facial nerve palsy in some vaccines [71,95]. In contrast, PC is a lipid that is also present in the surfactant of the airways, and due to its physiological properties, no adverse effects are expected [60,96]. We showed that the formulation of the inactivated vaccine with PC is necessary, as shown in Appendix A compared to Figure 4 [71]. It is worth mentioning that the vaccination of LEEI-RSV intramuscularly needed adjuvanting with Alhydrogel whereas the mucosal application only needed the formulation with PC.

In conclusion, this proof-of-principle study shows a novel potent method for the production of a mucosal inactivated whole virus vaccine formulated with PC. PC-LEEI-RSV protects against RSV by significantly reducing the viral load in vaccinated animals and presents a promising vaccine candidate. Further preclinical optimization and the clinical development of this vaccine candidate are still warranted.

## Figures and Tables

**Figure 1 viruses-15-01846-f001:**
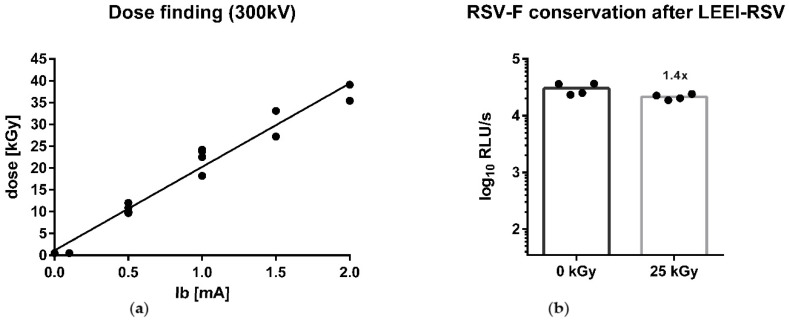
Dose finding and conservation of the RSV-F surface protein after irradiation. To determine the dose distribution, dosimetry of the low-energy electron irradiation (LEEI) was performed in the bag module with TTC. Two to three independent runs with different amperage at 300 kV are shown. The linear regression was calculated based on measured dosimetry (**a**). The conservation of RSV-F after LEEI was measured by an ELISA with the 18F12-antibody in the process control (0 kGy) and after inactivation with 25 kGy LEEI. Shown are the mean of each group and the fold reduction compared to 0 kGy (*n* = 4) (**b**).

**Figure 2 viruses-15-01846-f002:**
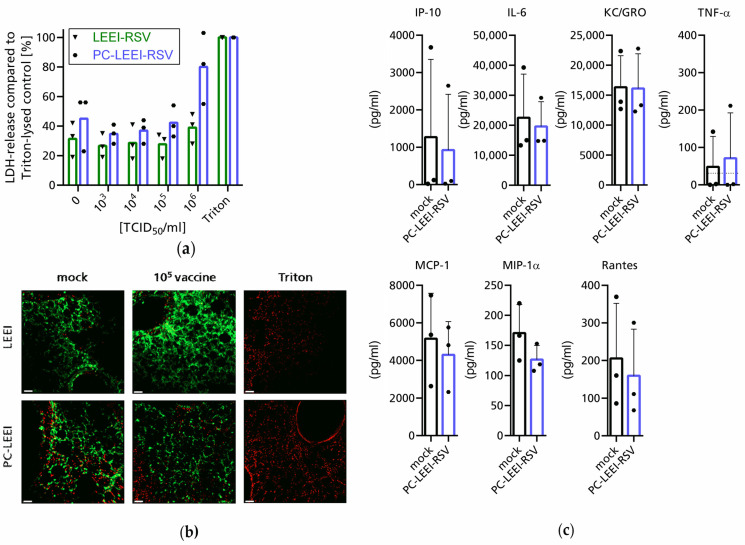
Evaluation of the adverse effects and toxicity of LEEI-RSV and PC-LEEI-RSV using precision cut lung slices (PCLSs). Murine PCLS were incubated for 24 h with different concentrations of LEEI-RSV or PC-LEEI-RSV medium (0 TCID50/mL or mock) or Triton (1% Triton X-100). LDH release after 24 h was measured and is shown as the percentage relative to Triton control (**a**). PCLSs were analyzed after 24 h incubation and stained with the fluorescence markers for living cells with calcein-AM (green) and for dead cells with Ethidium-homodimer (red) (**b**). Microscopic pictures of representative areas are shown (scale bar = 50 µm). After 24 h, the indicated cytokines were measured in culture supernatants at the concentration of 10^4^ TCID50 per reaction (MSD U-Plex assay) (**c**). Single dots indicate an independent experiment and bars the mean of all experiments (*n* = 3) with (**c**) or without (**a**) standard deviations; dotted line indicates the limit of detection.

**Figure 3 viruses-15-01846-f003:**
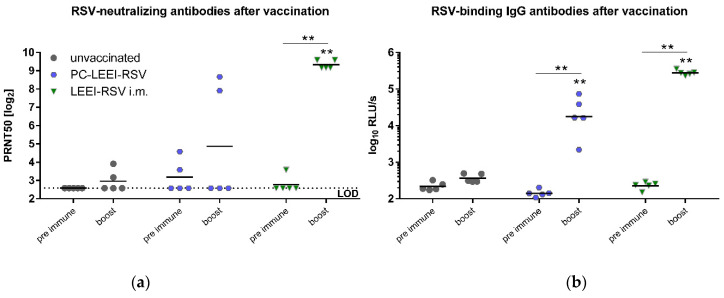
Systemic humoral immune response of the immunized animals. Mice were vaccinated in a homologous prime-boost regimen either with Alhydrogel adjuvanted LEEI-RSV intramuscularly (LEEI-RSV i.m.) or intranasally with PC-LEEI-RSV. Control animals were left unvaccinated. Before (pre-immune) and three weeks after the prime and boost vaccination, blood samples were collected to monitor the systemic humoral immune responses. The 50% plaque neutralization titers (PRNT50) in sera were tested in a microneutralization assay (**a**) and RSV-binding serum IgG antibodies (**b**) by ELISA. Every dot represents the mean of two separate measurements in duplicates of one animal (**a**). In (**b**), every dot is the mean of the duplicate of one animal and shown is a representative experiment out of two. Statistical evaluation performed by the Mann–Whitney test, either in comparison to the respective unvaccinated animal (indicated above each group) or in comparison to the group against the different timepoints (line) (**: *p* ≤ 0.01). (LOD = limit of detection for the virus neutralization titer at 1:6; *n* = 5). (relative light units per second = RLU/s).

**Figure 4 viruses-15-01846-f004:**
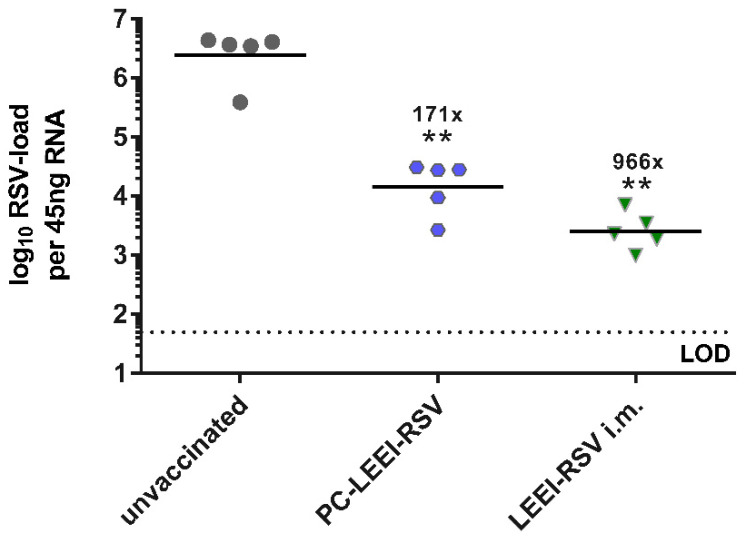
Viral load in the lungs after RSV challenge. BALB/c mice were vaccinated as described above. Four weeks after the boost immunization, animals were challenged with 10^6^ FFU RSV per mouse. RSV-load was measured five days after challenge in the lung via qRT-PCR. Shown is the viral copy number of RSV of each animal measured in duplicate with the corresponding geometric mean of each group. Calculated viral load reduction to the untreated control is presented. Statistical evaluation of the data was performed by the Mann–Whitney test in comparison to the untreated animal (**: *p* ≤ 0.01) (LOD = limit of detection at 50 FFU; *n* = 5).

## Data Availability

Data can be provided after request to the corresponding authors.

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
