# Peer review of "Mucosal Application of a Low-Energy Electron Inactivated Respiratory Syncytial Virus Vaccine Shows Protective Efficacy in an Animal Model"

_viruses, 2023, doi:10.3390/v15091846_

Round 1

Reviewer 1 Report

This is a very interesting study that provides information about the characterization of a novel inactivated vaccine candidate against human respiratory syncytial virus (RSV). This inactivated vaccine has been generated using the very promising methodology of low-energy electron irradiation (LEEI) as a safe, non-toxic and non-probe harming inactivation method.  Here, the authors evaluated safety, immunogenicity and efficacy of LEEI-inactivated RSV vaccine candidates (LEEI-RSV) in PCLS and Balb/c mice. A very interesting and novel part of this study includes the establishement of a mucosal application for the LEEI-RSV vaccine candidate by the use of newly established liposome phosphatidylcholine (PC) as adjuvant to provide a broad mucosal and systemic immune response after instillation. An important aspect of RSV vaccine development is to test the safety in vivo. For this they used different assays to evaluate on the activation of cytokine secretion in lung slices. In addition, the authors comparatively evaluated the LEEI-RSV vaccine candidate in mice using mucosal and intramuscular application route. Both applications resulted in the activation of robust humoral immune responses while the intramuscular application route seems to activate higher titers of specific antibodies. This is also confirmed for the efficacy testing. Here, the authors detected substantial lower titers of RSV in the lungs of mice that had been vaccinated intramusculary. Nevertheless, there was a profound effect of the mucosal PC-LEEI-RSV vaccination. The manuscript comprises an excellent introduction to the field summarizing the state of the art of the actual research. Overall, the authors describe a very interesting study that is of value to the field. My concerns are minor and relates to the activation of neutralizing antibodies (nAb) in the mice. The authors should include a section within the discussion explaining the titers of nAb in the control mice after the boost and in the PE-LEEI-RSV vaccination group on day 0. Moreover, did the authors also analyze the humoral immune responses after challenge infection? Is there an advantage of the mucosal application route? In addition, the authors should include a sentence within the discussion how the PE-LEEI-RSV mucosol application could be further improved. Finally, a section within the discussion also including the impact of cellular immune responses for the protective efficacy of RSV vaccines might further improve the manuscript.

Author Response

Reviewer 1:

This is a very interesting study that provides information about the characterization of a novel inactivated vaccine candidate against human respiratory syncytial virus (RSV). This inactivated vaccine has been generated using the very promising methodology of low-energy electron irradiation (LEEI) as a safe, non-toxic and non-probe harming inactivation method.  Here, the authors evaluated safety, immunogenicity and efficacy of LEEI-inactivated RSV vaccine candidates (LEEI-RSV) in PCLS and Balb/c mice. A very interesting and novel part of this study includes the establishement of a mucosal application for the LEEI-RSV vaccine candidate by the use of newly established liposome phosphatidylcholine (PC) as adjuvant to provide a broad mucosal and systemic immune response after instillation. An important aspect of RSV vaccine development is to test the safety in vivo. For this they used different assays to evaluate on the activation of cytokine secretion in lung slices. In addition, the authors comparatively evaluated the LEEI-RSV vaccine candidate in mice using mucosal and intramuscular application route. Both applications resulted in the activation of robust humoral immune responses while the intramuscular application route seems to activate higher titers of specific antibodies. This is also confirmed for the efficacy testing. Here, the authors detected substantial lower titers of RSV in the lungs of mice that had been vaccinated intramusculary. Nevertheless, there was a profound effect of the mucosal PC-LEEI-RSV vaccination. The manuscript comprises an excellent introduction to the field summarizing the state of the art of the actual research. Overall, the authors describe a very interesting study that is of value to the field.

My concerns are minor and relates to the activation of neutralizing antibodies (nAb) in the mice. The authors should include a section within the discussion explaining the titers of nAb in the control mice after the boost and in the PE-LEEI-RSV vaccination group on day 0.

Answer: The detected levels of neutralizing antibodies near baseline in the unvaccinated boost-sera and the PC-LEEI-RSV pre immune sera are more likely unspecific background. We included this sentence in the Discussion section. (Line 455-457).

Moreover, did the authors also analyze the humoral immune responses after challenge infection?

Answer: Thank you very much for this comment, we will consider it for the sampling in our next experiments. As we were focusing on the protective effects of the vaccines with respect to viral load and symptoms, we did not investigate antibodies after challenge.

Is there an advantage of the mucosal application route?

Answer: We described the mucosal application route so far (lines 538- 542), but we included further aspects to make the advantages clearer in the discussion.

Furthermore, it has already been observed that the induction of immune responses via a mucosal application induces include a broad mucosal immune response and systemic immunity depending on the vaccine candidate [30,31,97]. The intranasal application of an adenoviral vector vaccine against RSV lead to the induction of RSV-F specific CD8+ T cells, central memory CD8+ T cells and most importantly to tissue-resident memory CD8+ T cells in the lungs of vaccinated mice [30]. Similar observations were done with an in-tranasal RSV-F Nanovaccine inducing tissue-resident memory CD4+ and CD8+ T cells be-sides neutralizing antibodies [97]. These cellular immune response mechanisms are im-portant and necessary in a balanced and protective immunity against RSV [98]. (Line 526-535).

In addition, the authors should include a sentence within the discussion how the PE-LEEI-RSV mucosal application could be further improved.

Answer: Thank you for your suggestion. We added following sentences.

It is worth mentioning that the liposome-based formulation and the mucosal application can be improved further. For the intranasal application sophisticated devices are developed to produce size-specific droplets. However, the optimization of this mucosal vaccine candidate was beyond this proof of concept study. (Lines 465 - 468).

Finally, a section within the discussion also including the impact of cellular immune responses for the protective efficacy of RSV vaccines might further improve the manuscript.

Answer: We would like to thank the reviewer. We therefore added following paragraph into the Discussion section.

The intranasal application of an adenoviral vector vaccine against RSV lead to the induction of RSV-F specific CD8+ T cells, central memory CD8+ T cells and most importantly to tissue-resident memory CD8+ T cells in the lungs of vaccinated mice [30]. Similar observations were done with an intranasal RSV-F Nanovaccine inducing tissue-resident memory CD4+ and CD8+ T cells besides neutralizing antibodies [97]. However, the analysis of cellular immune responses for this proof of principle was beyond our study. (Lines 528-538).

Reviewer 2 Report

The manuscript by Everlein et al. demonstrates the efficacy of mucosally applied low-energy electron-irradiated RSV (LEEI-RSV) formulated with phosphatidylcholine-liposomes (PC-LEEI-RSV) both ex vivo (mouse lung slices) and in vivo (in mice). The experiments are well done and the results are impressive, but I have a significant problem about the originality of this paper, since it has massive similarity with several papers that this group of researchers has published over the past several years. Reference 41-44 of this group (with many common authors) together contain the same results and essentially all steps of the research, including inactivation of RSV by LEE, infection of mice, mucosal application of the LEEI virus, demonstration of antibody response and inhibition of the challenge virus. The overlap is particularly glaring in Reference 41 alone: "Automated application of low energy electron irradiation etc" Sci. Rep. 2020, 10, 12786, doi:10.1038/s41598-020-69347-7. The ONLY difference appears to be the use of phosphatidylcholine-liposomes. I would consider this a small, incremental difference, worth a very short note.

The authors need to improve on the originality and novelty of the current manuscript over references 41-44.

Much of the Discussion is repetition of the Introduction; Lines 511-521 are just one example.

English needs some improvement.

Quality is fine, but some places can be improved.

Author Response

Reviewer 2:

The manuscript by Everlein et al. demonstrates the efficacy of mucosally applied low-energy electron-irradiated RSV (LEEI-RSV) formulated with phosphatidylcholine-liposomes (PC-LEEI-RSV) both ex vivo (mouse lung slices) and in vivo (in mice). The experiments are well done and the results are impressive, but I have a significant problem about the originality of this paper, since it has massive similarity with several papers that this group of researchers has published over the past several years. Reference 41-44 of this group (with many common authors) together contain the same results and essentially all steps of the research, including inactivation of RSV by LEE, infection of mice, mucosal application of the LEEI virus, demonstration of antibody response and inhibition of the challenge virus. The overlap is particularly glaring in Reference 41 alone: "Automated application of low energy electron irradiation etc" Sci. Rep. 2020, 10, 12786, doi:10.1038/s41598-020-69347-7 . The ONLY difference appears to be the use of phosphatidylcholine-liposomes. I would consider this a small, incremental difference, worth a very short note.

The authors need to improve on the originality and novelty of the current manuscript over references 41-44.

Answer: We would like to thank the reviewer for the comments. The mucosal vaccination of LEEI inactivated viruses by an intranasal application was not tested in mice before. Therefore, we emphasise this novelty and originality compared to previous publications.

We described in our manuscript the application of the LEEI inactivated vaccine via the mucosal route: Discussion line 472f:

“The application route of LEEI inactivated vaccines was so far the intramuscular route but in this proof of concept study we observed that also the intranasal route induces protective efficacy based on the reduction of viral load.”

Indeed, the inactivated vaccine in the named publications was used, but the vaccine was not applied by the mucosal route. This was performed and described for the first time in the presented manuscript. Therefore, we highlighted this novelty and originality compared to the other publications in the manuscript as follows.

Line 101: In the present study, we investigated the mucosal application of LEEI-RSV for the first time.

Line 447: In this study we could show for the first time protection with the intranasal application of a PC-formulated LEEI-RSV in mice against RSV.

Line 462: These encouraging results demonstrate to our knowledge a proof of principle for a novel mucosal applied vaccine candidate which is inactivated by LEEI and formulated to improve immunogenicity and protective efficacy.

Much of the Discussion is repetition of the Introduction; Lines 511-521 are just one example.

Thank you for that point, we revised the Discussion section substantially, especially the part lines 511-521.

Round 2

Reviewer 2 Report

The authors have responded to my critiques satisfactorily.